# Electricity, Heat, and Gas Load Forecasting Based on Deep Multitask Learning in Industrial-Park Integrated Energy System

**DOI:** 10.3390/e22121355

**Published:** 2020-11-30

**Authors:** Linjuan Zhang, Jiaqi Shi, Lili Wang, Changqing Xu

**Affiliations:** 1State Grid Henan Economic Research Institute, Zhengzhou 450052, China; zlinj@163.com (L.Z.); wanglili11@ha.sgcc.com.cn (L.W.); xuchangqing@ha.sgcc.com.cn (C.X.); 2School of Electrical and Electronic Engineering, North China Electric Power University, Changping District, Beijing 102206, China

**Keywords:** industrial-park, integrated energy system, energy forecasting for electricity, heat and gas, deep learning, multitask

## Abstract

Different energy systems are closely connected with each other in industrial-park integrated energy system (IES). The energy demand forecasting has important impact on IES dispatching and planning. This paper proposes an approach of short-term energy forecasting for electricity, heat, and gas by employing deep multitask learning whose structure is constructed by deep belief network (DBN) and multitask regression layer. The DBN can extract abstract and effective characteristics in an unsupervised fashion, and the multitask regression layer above the DBN is used for supervised prediction. Then, subject to condition of practical demand and model integrity, the whole energy forecasting model is introduced, including preprocessing, normalization, input properties, training stage, and evaluating indicator. Finally, the validity of the algorithm and the accuracy of the energy forecasts for an industrial-park IES system are verified through the simulations using actual operating data from load system. The positive results turn out that the deep multitask learning has great prospects for load forecast.

## 1. Introduction

To date, massive use of fossil energy has caused serious harm for society, environment, and economy. At the same time, renewable energy has received widespread attention and become an important part of modern energy due to its good cleanliness and low carbon property. However, traditional research mainly focuses on the planning, design, and operation of individual energy systems, while ignoring the coupling between different types of energy source, resulting in greatly reduced flexibility in system scheduling.

Consequently, there is a need for energy systems to evolve from individual systems with little or no interdependencies into a complex set of integrated systems on large scales. On the basis, integrated energy system (IES) is proposed, which involved electricity, natural gas, and heat. The production, transmission, and consumption of the system can be optimized by IES as a whole, which is the key of the third industrial revolution [1]. At present, researches and works on IES can be divided into two characteristics based on application scenarios as [2,3] wide-area IES and industrial-park IES. Wide-area IES mainly includes the electricity transmission network and gas transmission network. Industrial-park IES involve the coordination of multiple regional energy systems on distribution network and can be seen as upgraded microgrids. The interaction between wide-area IES and industrial-park IES is shown in Figure 1.

Numerous functions in industrial-park IES will be considerably different compared with those in a conventional power grid, especially the load forecasting function. In recent past decades, the increasing quality of electrical load forecasts in short-term is significantly important for the accuracy and economics of marketers, which guides a lot of decisions of EMS [4]. With the wake of industry, energy forecasting will be much more significant than ever. As the interaction among a variety of energy sources become more relevant, the planning and optimization of IES will be confronted with many difficulties if there is no accurate energy forecast. Thus, efforts to allow load forecasts for diverse energy types to replace predictions of solely electrical load prediction will make an important impact on IES development.

There have been many researches involving load forecasting on electricity, including distribution network, microgrid, and even a nation. In addition, mature data mining machine learning is applied in prediction on power ranging from long-term to short-term horizon. Hong and Fan [5] offer a considerable review on load forecasting, from the perspective of feature use, methodologies and evaluation methods. A nonlinear combined estimator is applied to forecast short-term load in [6], which includes the whole chain of forecasting system, such as, preprocessing, forecasting model and the evaluation style. Ref. [7] suggests an innovative load forecasting model under cyberattack situation, which conduct operators to make suitable and appropriate dispatching decision. Ref. [8] proposes an improved multitask method for Bayesian Gaussian process by considering a wide variety of community’s energy consumption. Ref. [9] proposes the load prediction serving for day-ahead market, which is on account of hybrid artificial neural network, using an improved differential evolution algorithm as optimization approach. Apart from electric load forecasting, forecasting for other types of loads has also been addressed. Considering the property of energy source largely differs, the forecasting models are also built from diversified perspective. Ref. [10] presents predictions on volume of natural gas for day ahead by considering the gas change inertia, which is also faced with problems of gas dispatching issue. Ref. [11] deploys a linear regression to predict thermal loads 24 h ahead and compares the accuracy of each model. It can be concluded that forecasting the demand of gas and thermal in advance is also a vital preparation, which is also faced with dispatching, operation, and planning issue to cope with.

Most previously developed load forecasting methods suffer from two misunderstandings:

First, majority mature learning algorithms are of the shallow structure type. For example, in most of the neural network (NN) method, only a single hidden layer is included because of the lack of proper training algorithm for normal NN with various layers.

Therefore, complicated and error-prone hand-engineered features cannot be avoided in shallow approaches, and knowledge of specific domains is required for feature analysis. For example, maximum load, minimum load, daily load variation rate, and normalized mean square are the load characteristics extracted from the historical load data in [10].

Furthermore, different sources of energy demands are usually taken into account separately. The influence mechanisms of different load types exhibit complex relationships because of their different levels of relevance and independence, e.g., electricity and heat can be produced by a combined heat and power (CHP) system by consuming gas, electricity can be converted into heat through aground source heat pump, gas can be converted into heat through a gas-fired boiler, even further power-to-gas (P2G) functionalities can be widely used, and so on, in addition to centralized approaches to power, heat, and gas supply. The strong couplings among diversified energy loads in industrial-park IES have not yet been considered for load forecasting applications. On the other hand, the property of other energy source is dramatically different, especially for industrial use. As a result, how to make proper predictions becomes an important research topic.

Deep learning as a frontier of machine learning technology can enable the regression of complex nonlinear functions. The framework for deep learning is similar to the hierarchical structure of human brain, the core idea of deep learning algorithm is to, first, use an unsupervised learning method to extract data representation at a higher, more abstract level and, then, to use a supervised learning model for classification or regression. The differences between the widely used shallow learning algorithms and deep learning algorithms are as follows: (1) deep learning emphasizes the depth structure of model, which usually has 5 layers, 6 layers, 10 layers, or even more hidden layers and (2) deep learning also clearly highlights the importance of feature learning in building a multilevel, which means mapping relations from external low-level external signals to intricate high-level features during training layer by layer. Thus, the ability of multilayer to capture highly abstracted features from training data makes classification and regression easier. A structure comparison of shallow learning and deep learning is shown in Figure 2.

Several corporations, including Google, Tencent, IBM, and Microsoft, have devoted big data resource to researching this novel technology, and the development of deep learning have brought about great breakthrough in speech recognition, image processing, natural language, online advertising, and other fields in recent years. In 2011, deep learning technology was approved on speech recognition and the error rate was found to decrease by 20–30% [12]. In 2012, deep learning technology has also achieved significant success in the ImageNet competition [13] showing that the errors rate of this algorithm was 9% lower than that of classical strategies. In 2016, the DeepMind team from Google combined Monte Carlo random tree and reinforcement learning with deep learning to develop a new artificial intelligence in the Game of Go program named “AlphaGo,” which prevailed the human professional human ninth-dan player Lee Sedol with a total score of 4:1 [14]. Considering the effect of practical application, deep learning may be the most successful technology in the domain of artificial intelligence in recent decades. There have been a few publications concerning research on the employment of this promising technology in the field of energy systems. Refs. [15,16] report the application of a sophisticated deep learning technique for wind power projection. Ref. [17] proposes that deep learning can be applied to load prediction for short term. In addition, the simulation data is from electrical power consumption from city smart gird. Thus far, the applications of this promising technology in energy research have been very limited, without fully considering the advantage of autolearning feature from complex data space.

It is hoped that the remaining shortcomings of load forecasting methods for the IES context will be resolved. The framework of the article will be shown as addressing three respects of the problem.

The advancements in computer and sensor have generated a massive amount of data with high dimensionality in terms of features. This paper fully considers economic data, historical data, weather data, and calendar to enable them to be used to the maximum possible extent.Deep learning approaches have been effectively applied in energy demand forecasting research by highlighting the importance of feature learning in building model process, which means mapping relations from external low-level external signals to intricate high-level features during training layer by layer. These promising approaches have a feature abstraction capability that are appropriate for addressing the features without requiring careful engineering or extensive domain expertise.Different energy demands in IES always have complex interactions because of energy conversion and consumption fashion. Simultaneous learning for multiple relevant tasks can allow more precise energy demand forecasting results to be obtained. A deep learning architecture that also incorporates multitask learning is, therefore, proposed in this paper. It is fully demonstrated that multitask learning, for the simultaneous forecasting of electrical, gas, and heat loads, outperforms individually established prediction models. We investigate the effect compared with multiple instances of single-task to test the feasibility of multitask.

The framework of the article can be presented as follows. Section 2 introduces a basic theory on deep learning and multitask learning. Section 3 presents the proposed two-stage load forecasting model as well as the method used to supplement missing data, the principle applied for setting the inputs and the calculation of the accuracy index for evaluation. Section 4 describes the selection of the architecture parameters, the experimental results obtained using our approach, and the comparison with the results of other algorithms. In the end, Section 5 shows the conclusion and further work.

## 2. Fundamental Theory

### 2.1. Deep Belief Network

Deep learning simulates human and animal thinking, in which features of the input data are extracted from bottom to top, step by step. Deep belief network (DBN) based on the restricted Boltzmann machine (RBM) have been proposed as the basis of one widely applied type of deep learning method, at the bottom of such at model, an unsupervised DBN is employed to extract features from the data, whereas in the top layer, supervised regression is applied to generate the prediction results. The structure of such a deep learning regression forecasting model is illustrated in Figure 3.

DBN consists of lots of RBM stacks. The layers are the key components of RBM, including visible part and hidden part. The visible variables *vi* are linked with hidden variables *hj*. X represents input data. Y is a label that corresponds to the input data. *D* and *L* denote the hidden and the visible layer, respectively. *l_s_* represents the *s*th neuron of visible layer, and *d_t_* represents the *t*th neuron of hidden layer.

The model defines a probability distribution *R*(*l,d*) over *l* and *d* by an energy function *N*(*l*,*d*). The energy function *N*(*l*,*d*) is a general second-order polynomial and can be defined as follows:(1)$$N(l,d)=-\sum_{s=1}^{n_{l}}\sum_{t=1}^{n_{d}}w_{st}l_{s}d_{t}-\sum_{s=1}^{n_{l}}p_{s}l_{s}-\sum_{t=1}^{n_{d}}q_{t}d_{t}.$$

In an energy-based probabilistic models, the joint probability distribution can be defined as a function [18]:(2)$$R(l,d)=\frac{1}{Z}e^{-N(l,d)}$$
(3)$$Z=\sum_{s=1}^{n_{l}}\sum_{t=1}^{n_{d}}e^{-N(l,d)},$$
where *w_st_* is the weight between *s* and *t*. *q_t_* is bias of hidden unit *t,* and *p_s_* is the bias of visible unit *s*. The numbers of visible and hidden units are represented by *n_l_* and *n_d_*, respectively. *Z* is the normalizing factor; details of this function are introduced in [19].

When the visible layer is held fixed, output of hidden part is computed as:(4)$$p(d_{t}|l;\theta )=sigm(\sum_{s=1}^{n_{l}}w_{st}v_{s}+p_{t}).$$

When the hidden layer is fixed, output of visible part is calculated as:(5)$$p(l_{s}|d;\theta )=sigm(\sum_{t=1}^{n_{d}}w_{st}h_{t}+q_{s}),$$
where *sigm*() denotes activation function.

The training of an RBM falls under the purview of the field of unsupervised learning field [20], in which the parameters of a model are adjusted for training to complete the process of learning data characteristics. *L*(*θ*) as given below, is the log-likelihood function of the RBM, supposing there are *T* samples in dataset, where $l^{\left(k\right)}$ denotes the *k*th input sample. *θ* can be obtained through maximizing *M*(*θ*).
(6)$$\begin{array}{cc} M(\theta ) & =\sum_{k=1}^{K}\ln P(l^{\left(k\right)},d)\\ & =\sum_{k=1}^{K}(\ln\sum_{d}e^{\left[-N(l^{\left(k\right)},d)\right]}-\ln\sum_{l}\sum_{d}e^{\left[-N(l,d)\right]}) \end{array}$$
(7)$$\theta =\arg\max M(\theta )=\arg\max\sum_{k=1}^{K}\ln D(l^{\left(k\right)},d).$$

To obtain the optimal parameters, the gradient is defined as follows:(8)$$\begin{array}{cc} \frac{\delta M}{\delta \theta } & =\sum_{k=1}^{K}\frac{\delta }{\delta \theta }(\ln\sum_{n_{d}}e^{\left[-N(l^{\left(k\right)},d)\right]}-\ln\sum_{n_{l}}\sum_{n_{d}}e^{\left[-N(l,d)\right]})\\ & =\sum_{k=1}^{K}\left({\langle \frac{\delta (-N(l^{\left(k\right)},d))}{\delta \theta }\rangle }_{D(d|l^{\left(k\right)})}-{\langle \frac{\delta (-N(l,d))}{\delta \theta }\rangle }_{D(d,l)}\right) \end{array}$$

The details of the derivation formulas are provided in Ref. [21]. The notation <.>_P_ indicates the expectation value on distribution denoted by *D*. The first term in (8) is the expected value determined on the input data $l^{\left(k\right)}$. The second term in (8) is the expected value obtained for all possible inputs and hidden layer outputs. The approximate value of the second term in (8) can be received by alternating Gibbs sampling method [21,22]. Here, “*data*” and “*model*” are short for *D*(*d*|*l*^(*k*)^) and *P*(*l*, *d*), respectively, and the partial derivatives *θ* can be calculated as follows [23].
(9)$$\frac{\delta \ln M}{\delta w_{st}}=<l_{s}d_{t}{>}_{data}-<l_{s}d_{t}{>}_{model}$$
(10)$$\frac{\delta \ln M}{\delta p_{s}}=<l_{s}{>}_{data}-<l_{s}{>}_{model}$$
(11)$$\frac{\delta \ln M}{\delta q_{t}}=<d_{t}{>}_{data}-<d_{t}{>}_{model}.$$

In the top layer, we employ a back-propagation neural network (BP-NN) as a curve-fitting algorithm to simultaneously fine-tune the overall architecture parameters and predict the result. Because of the limited scope of this paper, no further discussion of the theory of BP-NN will be presented here. More details on the modeling process can be found in [20].

### 2.2. Multitask Learning

Multitask learning is classical branch in data mining and artificial intelligence [24]. The implementation of the same network to perform classification and regression for multiple tasks provides the opportunity for multitask learning. During the learning procedure for one task, information on other tasks can be included in parallel via shared weights to improve the learning accuracy. This is a classic paradigm in machine learning. A comparison of multitask and single-task architectures is shown in Figure 4.

In an industrial-park IES, various energy systems are closely connected to each other in the form of energy flows. A considerable amount of information is shared among different energy carriers according to energy transformation in IES, as mentioned in Section 1. Therefore, it seems promising to incorporate multitask learning into our architecture. In a single-task learning approach, each regression approach is always separately trained for a different task, each energy source is predicted separately. In a multitask model architecture, we put a variety of related objects together in regression level. The included tasks are parameters fine-tuned through back-propagation. Moreover, the sharing mechanism in the DBN will also elaborate the features initially produced by these deep layers, thereby allowing the subsequent joint fine-tuning to better represent the complex correlations among different energy carriers and improving the generalization performance. Thus, the information from one task can aid in more effective learning for related tasks, allowing improvements to be achieved through weighted sharing.

## 3. Model

In deregulation environment, it is necessary to implement load forecasting function for energy marketers, load aggregators, and independent system operator (ISO). In addition, many of these agents may lack computing experience and access to high-performance servers. The investigation of such scenarios will require the development of a universal model that is suitable for application by a wider variety of entities. Based on the relevant application demands and algorithm requirements, we propose a two-stage short-term hourly load projection for multiple energy types. The method takes the idea of a deep multitask learning framework composed of modules for off-line training and on-line prediction. whose structure is shown in Figure 5.

Off-line training plays an important part in guaranteeing load forecasting accuracy, which is essential for parameter optimization. The off-line segment collects load data through wide cyber-physical system to achieve optimal parameters of algorithm. According to the computational burden and data timeliness, each training dataset used in off-line training needs to span the same appropriate fixed time range. Since a deep learning algorithm requires considerable calculations and storage resources because of its special structures, which contain enormous numbers of nodes and layers compared with a conventional shallow algorithm, services related to this portion of the framework could be provided by industrial-park IES service providers or the operators of the energy distribution systems in practical application.

Meanwhile, the on-line prediction module can be configured for execution on normal computers or embedded systems. Once the parameters of the algorithm are found through off-line training, the trained program can be downloaded to a normal PC to implement the real-time portion of the algorithm. Real-time data can be metered by smart terminal, which means true previous historical load is used as input of on-line prediction.

Load is a nonstationary process due to utility growth, variations in weather or seasonal changes. Hence, a periodic updating mechanism that is sensitive to recent load trends can assist in producing better results. When the error is larger than a certain error threshold over a certain period of time or when the same parameter has been applied over a fixed time threshold, an update strategy will be triggered that will cause a new data sample to be used to reobtain the model parameters, thus serving as a rolling updating mechanism.

### 3.1. Preprocessing for loads Data and Normalization

Data preprocessing or data cleansing always plays a significant role in load forecasting. Influenced by factors such as measurement equipment characteristics, grid faults and energy supply constraints, the historical data collected by data acquisition devices can be expected to be subject to problems such as missing data and abnormal fluctuations. Therefore, these data need to be supplemented or revised before being used. Load data of the same type and from the same date are selected for supplementation or correction using the weighted average method:(12)$$x_{i}=\lambda_{1}x_{i-24}+\lambda_{2}x_{i+24},$$
where *x_i_* represents the load value at the ith sampling point, *λ*_1_ and *λ*_2_ are the weights used for the calculation (both could be 0.5), and *x*_*i*−24_ and *x*_*i*+24_ represent the load values at the same time on the previous day and the next day, respectively. Finally, the historical load data series should be normalized to the interval [0, 1] as follows:(13)$$y_{i}=\frac{x_{i}-\min(x)}{\max(x)-\min(x)}.$$

### 3.2. Input Properties Setting

The input variables can make maximum use of the information provided by cyber-physical systems and numerical weather forecasts. The input features include historical load data, weather data, calendar rule, and economic data. The historical load data comprise historical datasets of heat load, gas load, and electrical load. The weather data consist of temperature, humidity, and wind speed. Weekdays and holidays (weekends and public holidays) are classified by calendar rule in accordance with the public holidays observed in China. In addition, for economic data, depending on the size of the region, one can use regional GDP data, the stock prices of corporations, or other financial data. Considering that the industrial park belongs to listed company, we also add some economical index into dataset, such as stock price. The input variables can make maximum use of the information provided by cyber-physical systems and numerical weather forecasts. The setting of input properties is shown as Figure 6.

### 3.3. Evaluating Indicator

Assume *r* is the total quantity of days. Let *a*(*k*) represents the *k*th actual value, and let *c*(*k*) represents corresponding forecasted value. Then, the MAPE and the corresponding mean accuracy (*MA*) are defined as follows:(14)$$MAPE=\frac{1}{r}\sum_{k=1}^{r}\left|\frac{a(k)-c(k)}{a(k)}\right|\times 100\%$$
MA = 1 − MAPE.(15)

Given that various sorts of outputs are predicted by multitask model, we use the weighted mean accuracy (WMA) to evaluate the whole model performance, which is calculated using the different load forecasting errors as weights. α*_k_* represents the weight corresponding to *MA_k_*, which is the mean accuracy of the *k*th sample.
(16)$$WMA=\alpha_{1}MA_{1}..+\alpha_{k}MA_{k}.$$

Here, it is implied that the type of energy load on which we are most focused is that with the highest precision. Because of the dominant importance of electricity among the various energy types in the IES context, the electrical load forecasts must be given a higher weight than the other tasks in this paper. We set weights of electricity, gas, and heat as 0.6, 0.2, and 0.2, respectively.

## 4. Experiments and Results

The experimental dataset is collected from the industrial-park IES demonstration project of Goldwind Technology Co., Ltd., in Daxing District, Beijing, China, consisting by electricity, natural gas, and heat. The interactions among the various energy carriers in this industrial-park IES demonstration project are shown in Figure 7. CHP, electric boiler, and gas-fired boiler are established as energy-conversion facilities to meet the demands for transformation among various energy sources. The industrial-park IES is configured by storage station to develop economic benefit as a whole. The PV and wind generator are also configured in IES to utilize the renewable energy resource. It is noteworthy that although industrial-park IES includes the industrial production module, the community also hosts office part and residential zones. Energy loads in different period of year are shown in Figure 8.

Taking into account of seasonal fluctuation of energy load in different periods of year, two patterns of data that represent energy consumption of summer and winter are used in our experiment. The sampling interval time is 1 h. The task is to forecast the electricity, heat, and gas load of next hour.

Pattern 1 (winter): The dataset of the period of January 2014 to December 2015 is applied for training, and the dataset in January 2016 is used as test data.

Pattern 2 (summer): The dataset of the period of July 2014 to June 2016 is used as training data, and the dataset in July 2016 is used as test data.

### 4.1. Parameters Selection of Deep Multitask Learning Architecture

To ensure the optimal structure of the deep multitask learning algorithm to obtain results with the minimum prediction error, it is particularly important to properly choose the exact number of layers, nodes, the training iterations, and other relevant parameters. The improper framework of model leads to negative effect for model performance. In our case, data in Pattern 1 is used as experimental data for the off-line training module. These parameters are selected using the longitudinal comparison method. When testing the effect of a given parameter on the prediction results, the other parameters are held fixed. The weights number and the training time are deemed representative of the spatiotemporal complexity of the problem.

The influence of the number of layers on the WMA is presented in Table 1. The number of nodes in one layer can be held fixed (256 nodes). As number of layers is increased from 2 to 4, the WMA is gradually improved. When the number of layers reached 4, the WMA of the test dataset is 0.9563, and the optimal number is achieved. When it is beyond 4, the WMA is decreased because the model becomes too complicated, leading to “over-fitting” and causing the spatial and temporal complexity to simultaneously increase.

The influence of the quantity of nodes in one layer on the WMA is presented in Table 2. The number of layers was held fixed (4 layers). As the quantity of nodes is increased from 64 to 256, the WMA increases proportionally. This suggests that if we continue to reduce nodes in each layer, the model may be unable to learn representative features. When the number of nodes per layer was 256, the WMA reached its optimal value of 0.9563; as the number of nodes rise to more than 256, the additional improvement in precision become negligible, whereas the calculation time greatly increased. Thus, including more nodes will impose an unnecessary burden for model training.

When the model is trained using fewer than 600 iterations, the WMA rises for both datasets with the increase in the training iterations. When it exceeds 600, WMA for training dataset continues to increase. However, the WMA for the test dataset begins to show a continuous decline, which is also characteristic of the over-fitting phenomenon. Therefore, the best architecture parameters are found to be 4 layers, 256 nodes in each layer, and 600 training iterations.

### 4.2. Various Types of Load Forecasts Results

Taking into account of seasonal fluctuation of energy load in different period of year, the real load samples and forecasting data for 1 week in January or July 2016 are chosen to analyze the forecasting results and errors distribution. The time ranged from the 72nd h to the 120th h corresponds to holidays, and the remaining time corresponds to working days. The prediction results for each type of load and the hourly MAPE variations for the different types of loads are depicted in Figure 9 and Figure 10.

The curve of energy load appears to reflect seasonal fluctuation in different seasons. In summer, due to high temperature, a large number of air conditioners and electric fan device are put into operation. Electrical load appears to be obviously increased, which makes peak electricity consumption occur frequently. Meanwhile, the demand for heat is limited except some necessary requirements on industrial production. On the contrary, in winter, the demand for gas or heat load is greatly increased because of people’s living and industrial use needs.It can be concluded from Figure 9d and Figure 10d that the accuracy is high in the rising and falling regions of the load curves. The largest errors typically appear closely the maximum and minimum points of daily loads. Moreover, the general trends for the different types of loads largely coincide regarding on-peak and off-peak periods.From Figure 9 and Figure 10, we can see that the fluctuations in electric load are more severe than those in the other loads during all periods. This is because the inertia of the heat and gas loads is greater than that of the electrical load, and the diversified nature of the storage of gas and heat ensures sufficient reserves for operation. Thus, the gas and heat load forecasts have less error. The MAPEs of different pattern for the electrical load, gas load, and heat load are shown in Table 3.Although holidays and weekdays are clearly classified by calendar rules in the input settings, the forecasting error for holidays still appears to be larger than that for weekdays. This error mainly originates from the uncertainty in people’s activities on holidays (relaxation, entertainment, etc.). Moreover, enthusiasm for work or production may either increase or decrease on the days surrounding holidays, which lack the regularity of normal weekdays. The average load MAPEs for holidays, the days surrounding holidays, and weekdays are 6.89%, 4.64%, and 2.79%, respectively.Because of its limited capacity, forecasting for industrial-park IES is more complicated to implement than forecasting for a large power grid because of the higher randomness in the historical load curves. The fuzziness of the loads is largely reflected in the load forecasting result, and the random errors cannot offset each other for such a small-scale system. Therefore, this situation will certainly lead to larger fluctuations.

### 4.3. Comparison of Multiple Forecasting Methods

In this section, several commonly used data mining methods are applied as benchmark to compare with deep learning, which cope with same multitask load forecasting regression problems, e.g., an ARIMA, support vector regression (SVR), gradient boosted trees (GDT), and BP-NN. The dataset differed slightly because of the different conditions for each method. The results for different methods are compared in Table 4.

The conclusion can be drawn that deep learning is superior in terms of generalization ability and learning capacity, because the process of unsupervised learning allows intrinsic relationships among a variety of factors to be found. The ARIMA model is limited by its algorithm structure such that it can only use historical load datasets and cannot effectively account for the impact of other factors. The forecasting results of the BP-NN and SVR models also show a gap with respect to the deep learning results because their restricted learning capabilities impose limits on their ability to handle complex data models.

In addition, when the shallow algorithms (such as, BP-NN and SVR) are used to make prediction of the electricity, heat, and gas, the forecasting error of thermal and gas is obviously lower compared with electrical power. The reason is that the usage of gas and heat profile are more regular, whose change is mainly from production arrangement of industry, and electrical load always has serious volatility due to nonstorable characteristic. When deep learning method is deployed to forecast the different source of load, the forecasting error gap between electrical load and other tasks is reduced, meanwhile, the prediction accuracy for all tasks has been improved, which indicates that the unsupervised learning part can learn the better features provided by other tasks by weight sharing of multitask learning. The task of electrical load forecasting is able to learn knowledge from the gas or heat profile to aid improve the forecasting performance.

The following comparative analysis focuses on the details of the underlying advantages of the deep learning algorithm compared with the shallow learning of BP-NN, the main difference between deep learning and BP-NN is whether including unsupervised learning or not as pretraining method in modeling process. Thus, we design the experiment concerning about the loads forecasting model with or without the unsupervised training to test network error for both methods.

The conclusions of experiment can be drawn as follows from Figure 11. The forecasting error between traditional BP-NN and DBN is basically the same concerning the structure with one single hidden layer. The effect of BP-NN is very unsatisfactory as the number of layer increases, and network error becomes large. The reason is that gradient explosion existence when the error signal is transformed from top layer to bottom layer in BP-NN, resulting in inefficient training of the parameters at the bottom of the network, which is corresponding to bad generalization performance. On the other hand, BP-NN parameters are initialized randomly, and weights and bias of BP-NN are adjusted in iteration process until convergence. Hence, the starting point of iteration is far away from the optimal region and the loss function always contains multiple local minima. However, unsupervised training is implemented before supervised learning process in DBN, their weights are initialized by DBN instead of random generation. Then, DBN is expanded into a BP neural network, whose network is optimized by BP, to overcome the shortcomings of easily falling into local optimum and long training time.

### 4.4. Comparison of Different Training Models

In this section, the effectiveness of multitask learning is validated. The deep architecture network with multitask aggregated at different level is trained, and we compare the results with those of training for each task separately. The comparisons of single-task and multitask learning are all processed using the same deep learning architecture. The same training data are also used to support the different learning approaches to ensure the fairness of the experiment.

Figure 12 compares the multitask results with single-task results. The prediction accuracy of three kinds of loads is significantly improved by jointly training tasks. We can see that the MA for forecasting electricity-gas together is less than those of other multitask aggregated style for the reason that electrical flow and gas flow may have weak coupling with each other. On the contrary, complicated and diversified transformed fashion between gas flow- and heat flow-enabled multitask for gas-heat obtains relatively better forecasting results. In addition, compared with single-task, the accuracy improvement of each task for training all loads reached approximately 2.7%, which has the best performance in all multitask models. Instead of forecasting each load separately, the multitask method analyzes the complex coupling relationship among several types of input information, meaning that the training task for each type of load considers information related to the other load types. The multitask for all loads leads to a largest reduction in error, which effectively improves the accuracy of the results.

Moreover, multitask exhibits great performance in terms of the computational cost, which is always a significant criterion for evaluating a model. In Table 5, it is clearly seen that although the single-task training for each individual energy source requires a relatively short amount of time compared with the multitask training, the training process for multitask learning requires less time than the sum of all training times for the single-task training processes. Therefore, simultaneous training for multiple related tasks is more convenient because it only needs one training run rather than requiring the training process to be executed separately for each task.

## 5. Conclusions

A multitask load forecasting is proposed based on deep learning approach for multiple types of energy sources in an industrial-park IES. Through a case study, we demonstrate that electricity, heat, and gas load always have its own characteristics at different periods, and deep learning offers superior accuracy of energy load prediction compared with other shallow learning algorithm because of unsupervised learning method, we also show that simultaneous training for multiple related tasks has good performance, which can effectively solve the problems related to training approximation and training duration.

Much further study is still required concerning deep learning in IES load prediction research.

Considering the symmetric structure of the RBM model, there are full interactions between the visible and hidden layers. Consequently, the model possesses large numbers of weights and biases that must be computed. During the off-line training process, the host computer or server is always subjected to massive burdens of storage and calculation. As the next step of research, GPU computing or distributed optimization can be adopted to improve the computing efficiency to allow the optimal solution to be obtained in a shorter period of time.In the deep learning algorithm, the optimal numbers of nodes and layers are selected through observation (trial and error approach), which makes the modeling process complicated and inconvenient. A new method of tuning the optimal architecture parameters should be sought to improve the feasibility of the algorithm.The IES concept involves integrating multiple energy subsystems to achieve beneficial effects. Accordingly, the further integration of additional types of energy is inevitable to extend the scope of IES operation. Water loads, cooling loads, and other forms of energy loads are also common kinds of loads that can be considered in future load forecasting efforts.

## Figures and Tables

**Figure 1 entropy-22-01355-f001:**
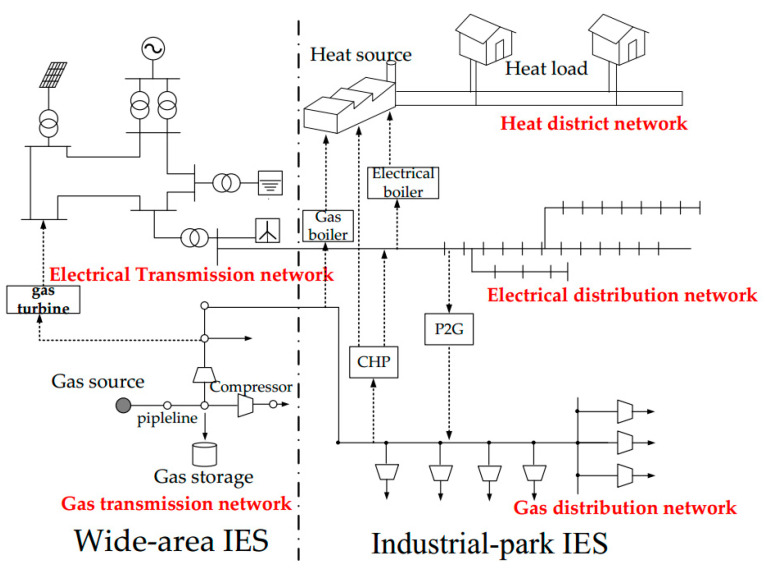
Wide-area integrated energy system (IES) and industrial-park IES.

**Figure 2 entropy-22-01355-f002:**
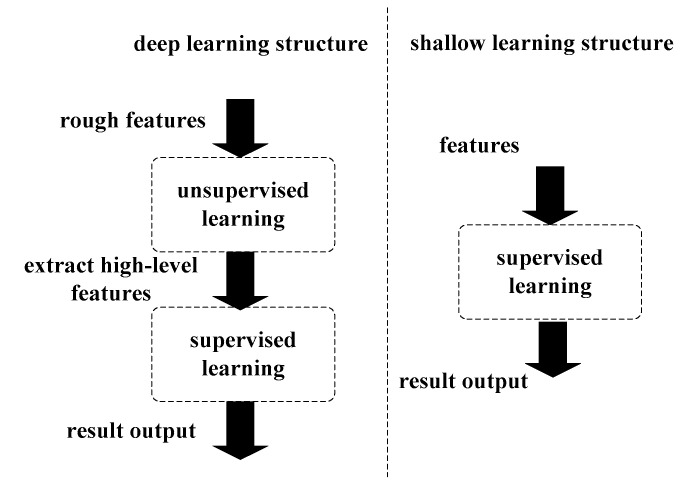
Comparison of shallow learning and deep learning structure.

**Figure 3 entropy-22-01355-f003:**
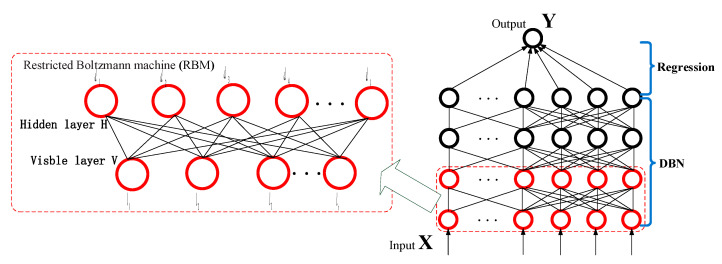
Deep learning architecture for forecasting regression.

**Figure 4 entropy-22-01355-f004:**
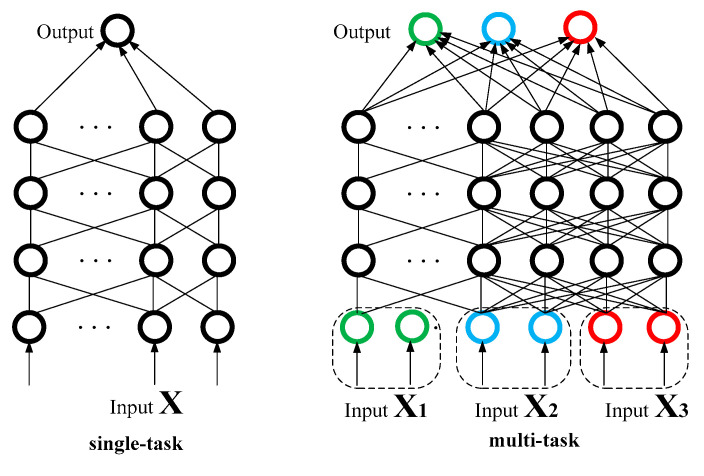
Multitask and single-task.

**Figure 5 entropy-22-01355-f005:**
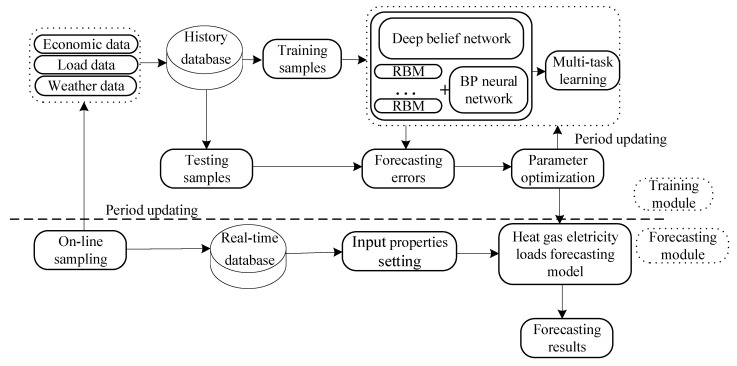
Framework of energy forecasting model in industrial-park IES.

**Figure 6 entropy-22-01355-f006:**
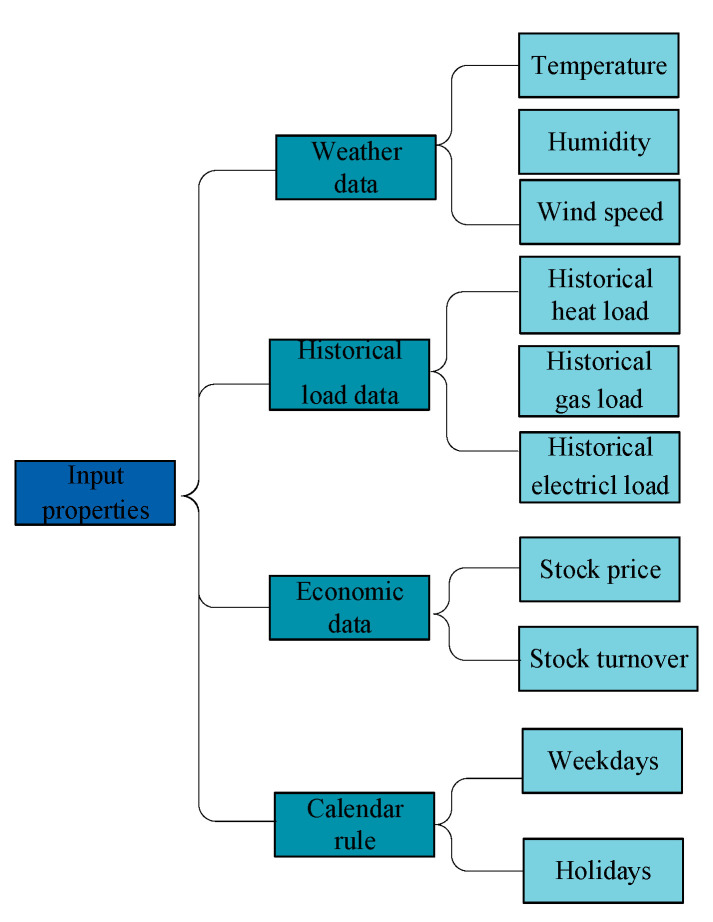
Setting of input properties.

**Figure 7 entropy-22-01355-f007:**
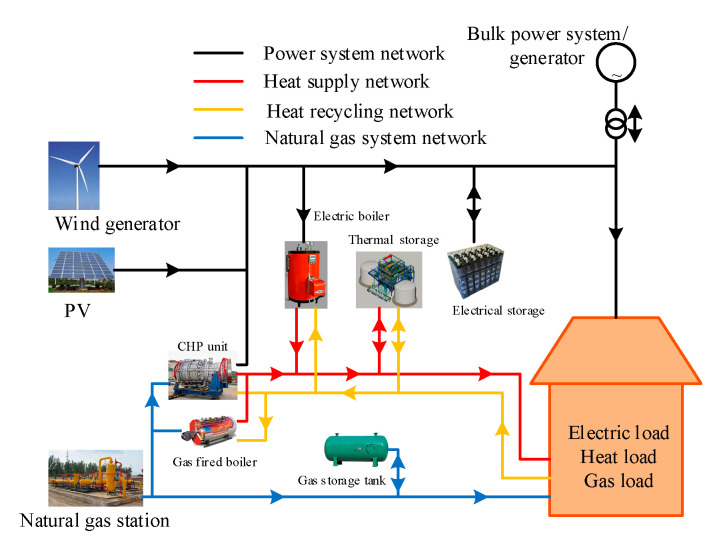
The interactions among various energy carriers in industrial-park IES.

**Figure 8 entropy-22-01355-f008:**
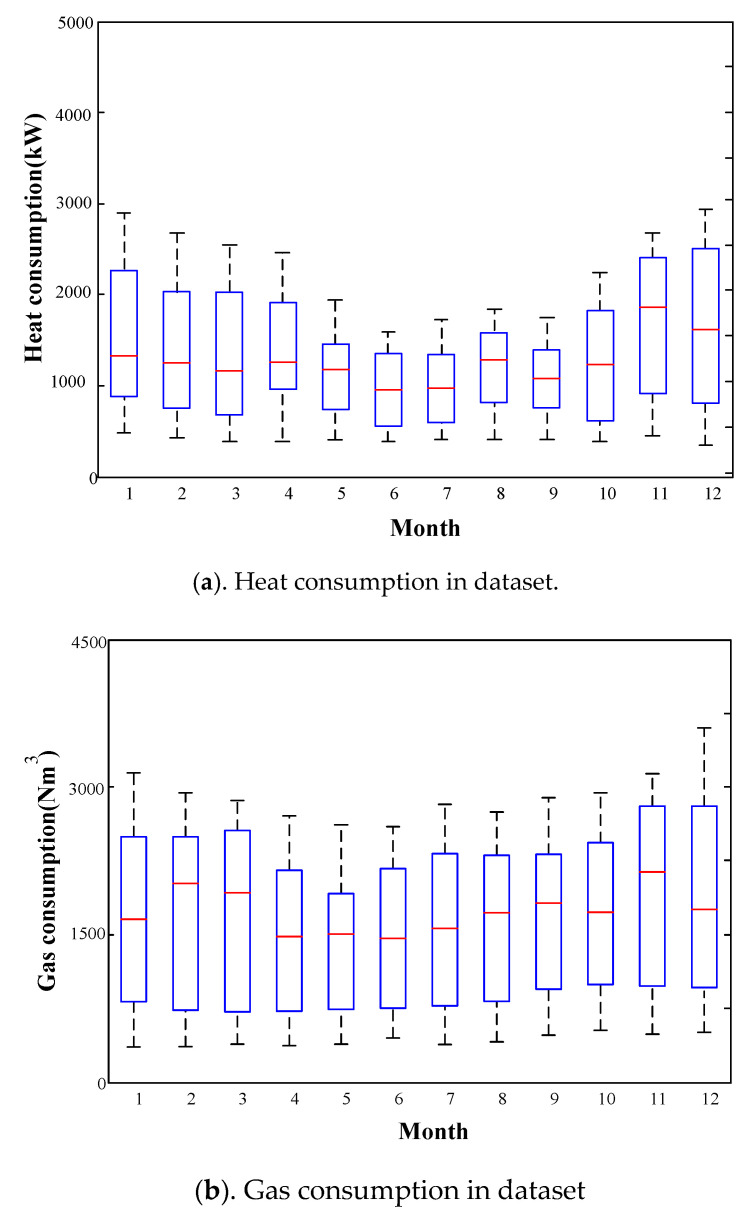
Energy loads in different period of year.

**Figure 9 entropy-22-01355-f009:**
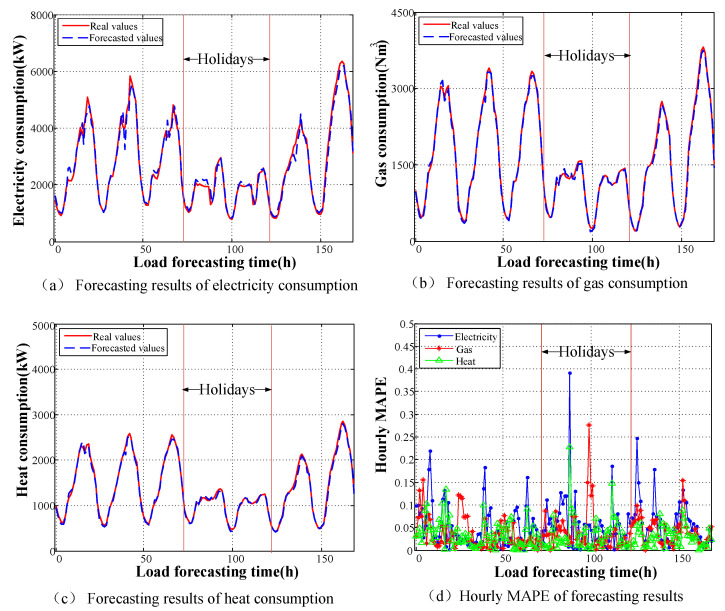
Forecasting results of energy consumption in summer.

**Figure 10 entropy-22-01355-f010:**
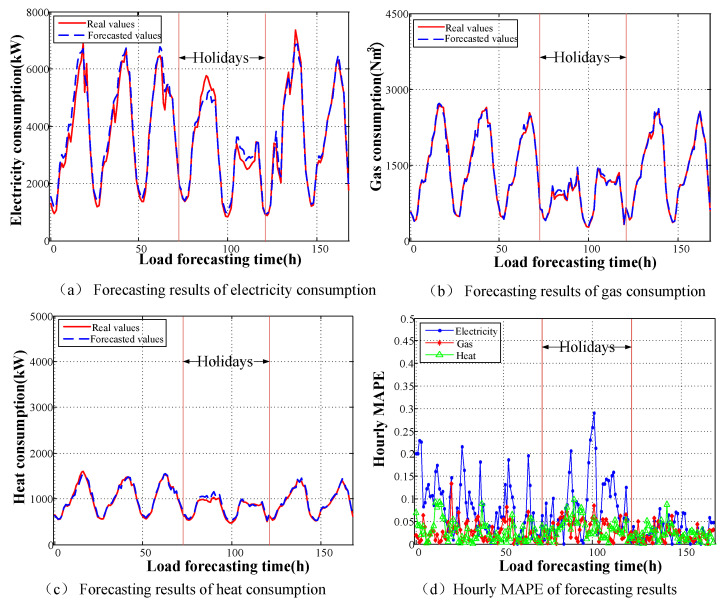
Forecasting results of energy consumption in winter.

**Figure 11 entropy-22-01355-f011:**
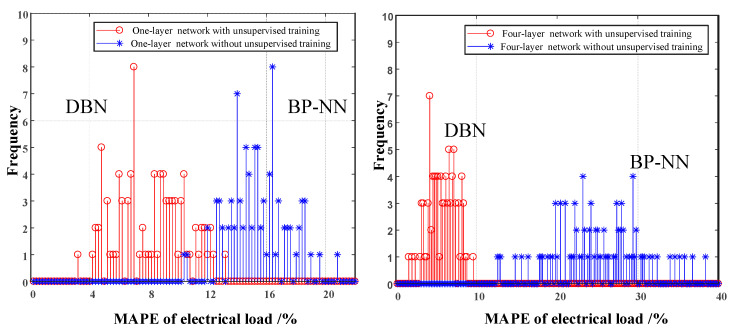
Error of the network with or without considering unsupervised training.

**Figure 12 entropy-22-01355-f012:**
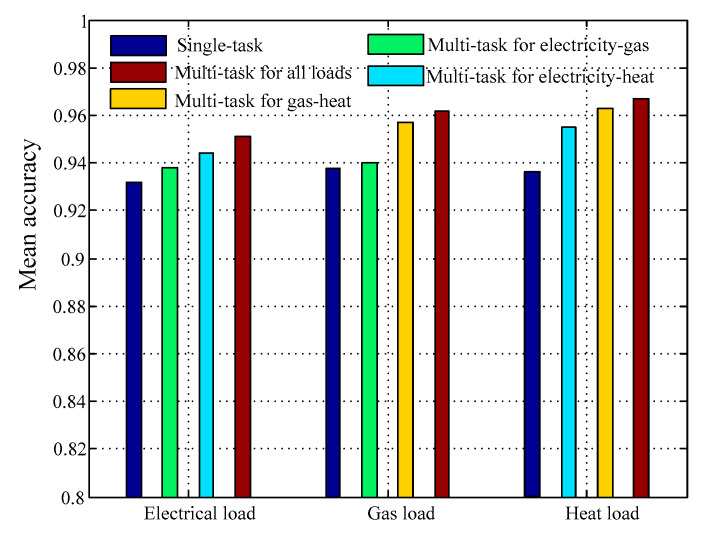
Comparison of multitask and single-task results based on test dataset.

**Table 1 entropy-22-01355-t001:** The comparison of layer number.

Layers	WMA	Number of Weights	Training Time/s
1	0.8991	140,032	1654
2	0.9281	205,568	2363
3	0.9484	271,104	3091
4	0.9563	336,640	3950
5	0.9501	402,176	4770
6	0.9432	467,712	5554

**Table 2 entropy-22-01355-t002:** The comparison of node number in single layer.

Nodes in Single Layer	WMA	Number of Weights	Training Time/s
32	0.858	13,408	153
64	0.901	35,008	425
128	0.9321	102,784	1181
256	0.9563	336,640	3950
512	0.9566	1,197,568	14,063
1024	0.9396	4,492,288	53,063

**Table 3 entropy-22-01355-t003:** The comparison of forecasting error in different dataset pattern.

Load Type	MAPE In Pattern 1	MAPE In Pattern 2
Electrical load	4.91%	6.25%
Gas load	3.83%	2.88%
Heat load	3.28%	2.16%

**Table 4 entropy-22-01355-t004:** Comparison of multiple prediction methods.

Algorithm	WMA	MA of Electric Load	MA of Gas Load	MA of Heat Load
DBN	0.9563	0.9509	0.9617	0.9672
SVR	0.9179	0.9012	0.9383	0.9477
GDT	0.9102	0.8922	0.9239	0.9450
BP-NN	0.9004	0.8889	0.9009	0.9342
ARMIA	0.8589	0.8476	0.8654	0.8863

**Table 5 entropy-22-01355-t005:** Comparison of the training time in different training modes.

Training Mode		Training Time
Multitask for all loads		3950 s
Multitask for electricity-gas		3670 s
Multitask for gas-heat		3710 s
Multitask for electricity-gas		3666 s
Single-task	Electricity	3160 s
	Gas	3099 s
	Heat	3117 s
	Sum	9376 s

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
