# Peer review of "Electricity, Heat, and Gas Load Forecasting Based on Deep Multitask Learning in Industrial-Park Integrated Energy System"

_entropy, 2020, doi:10.3390/e22121355_

Round 1

Reviewer 1 Report

The manuscript presents a forecasting strategy applied to an industrial-park integrating several energy vectors (as electricity, heat and gas). The novelty seems to rely on the use of deep learning architectures to incorporate multi-task learning.

The research reveals to be interesting to this domain with some efforts to demonstrate its effectiveness.

Nevertheless, some attention should be paid to clarify some doubtful items:

   -  ISO is always used as abbreviation, but there is no concern to present the abbreviation (Independent System Operator??) the first time it appears.

   - in section 3.2, when MAPE is presented as evaluation indicator, does it express consumptions on a daily basis? In this case, the daily consumption? Why couldn't be considered a higher resolution? (mainly when forecasting electricity consumptions); By the way, load forecasting results shown in section 4.2 present higher resolution.

   - it is expressed that WMA (used as weighted mean accuracy) is based on  the weighted relative importance of electricity, gas and heat (with weights chosen as 0.6, 0.2 and 0.2, respectively). How these specific weights were chosen, and what would be the influence of changing these weights? Have the authors tried different weights combinations? A sensitivity analysis would enhance the quality of this analysis;

  - when the different patterns of data (winter and summer) are described, it seems that there exist time discontinuities in-between, but not easily perceived. For example, winter dataset occurs between January 2014 to December 2015 (for training data) and from january 2016 (as test data) until ??? (no information about the whole data available). But, in-between this winter pattern, summer periods take place, so it is hard to understand the distinction between winter and summer periods;

 - when results are presented in Tables 1 and 2, no information is given about the number of simulations  that were considered. If it results from a single simulation, the resulting accuracy can be seriously dependent on the initial parameters (weights and biases);

  - if pattern 1 corresponds to winter period and pattern 2 corresponds to summer period, why do the authors do not consider the same sequence when showing Figures 7 and 8?

  - in Section 4.4, it was proposed ARIMA (and not ARMIA) as algorithm, and it is written that this model is limited because it only considers historical load datasets. I agree with the limitations of this model when compared with the concurrent ones, but an alternative model could be used (such as ARMAX or ARIMAX) to accomodate external (exogenous) inputs. 

A serious English revision should be made. Some examples of a poor quality written English:

    - in abstract (line 16) it is written: "... the whole energy forecasting model are introduced". It shouldn't be in plural, as it is a single model that comprehends different energy vectors at a once. 

    - in the end of Introduction it is written "... Section V shows the conclusion and plans". Plans shoud be changed to further work or further research.

    - several sentences are not initialized by capital letter - example line 199 of Section 3.

   - in the begining of Section 3.2, should be "Let a(k) represents the kth actual value. and let c(k) represents the corresponding forecasted value".

  - some sentences seems to be incomplete or do not make sense: some examples: "section 4, after figure 6 or the first sentence of section 4.2, or even the first sentence of third paragraph of section 4.4 (line 351);

   - the relative pronoun "who" is not suitable to be used in Section 4.4 (line 373) 

 - the references style needs to be normalized;

Reviewer 2 Report

36, 37: You need to define what a wide-area IES is and clarify the distinction between a wide-area IES and an industrial-park IES

52-70: Are there any significant connections between the techniques used in the journal papers cited and your approach?

72: You need to give a clearer explanation of "shallow structure". Do you simply mean neural networks with at most one hidden layer?

89-107: You mention the success of deep learning in many different areas that are only remotely connected to load prediction. Why would the success of deep learning in playing the Game of Go be relevant to load prediction? 

113-118: You mention: "Deep learning approaches have been effectively applied in energy demand forecasting" Can you cite any references? Such references would seem appropriate in lines 89-107

120: The distinction between single-task and multi-task learning is not clear at this point in the paper. An example would be helpful.

134: It would be helpful to provide a verbal description of Deep Belief Networks.

135: It is not clear what RBM refers to until Figure 2

189: In Section 3, a detailed description of how the inputs are fed into and processed by the DBN/Regression is needed. Did you use a particular implementation of DBN and if so, which implementation?

219: The diagram in Figure 4 is helpful

223: It is not clear what "calendar rule" means.

221-229: Was there any scaling performed on the inputs? It would be helpful to provide units for the meteorological and load data variables. Also, a sample table of values would be helpful especially for the meteorological and load data variables.

229: stocking price => stock price

246: Some additional explanation mentioning the distinction between the yellow and red lines in Figure 5 would be helpful

332: In looking at the plots in Figures 7, 8, it doesn't seem that the forecast error grows significantly as the forecasting lead hour increases. This seems curious. If you increase the forecast window to 240 hours, do you see similar performance? Were the meteorological variables helpful? If removed, how would the load prediction suffer? Which input variables were the most significant?

342: Gradient boosted trees would be interesting to include in your mix of alternate prediction methods.

388: Figure 10 is helpful
